# Enhancing Prediction for Tumor Pathologic Response to Neoadjuvant Immunochemotherapy in Locally Advanced Esophageal Cancer by Dynamic Parameters from Clinical Assessments

**DOI:** 10.3390/cancers15174377

**Published:** 2023-09-01

**Authors:** Xin-Yun Song, Jun Liu, Hong-Xuan Li, Xu-Wei Cai, Zhi-Gang Li, Yu-Chen Su, Yue Li, Xiao-Huan Dong, Wen Yu, Xiao-Long Fu

**Affiliations:** 1Department of Radiation Oncology, Shanghai Chest Hospital, School of Medicine, Shanghai Jiao Tong University, Shanghai 200030, China; swuing@sjtu.edu.cn (X.-Y.S.);; 2School of Medicine, Shanghai Jiao Tong University, Shanghai 200025, China; 3Department of Thoracic Surgery, Shanghai Chest Hospital, School of Medicine, Shanghai Jiao Tong University, Shanghai 200030, China

**Keywords:** neoadjuvant immunochemotherapy, organ preservation, esophageal cancer, CT, esophagogram, esophagoscope, response to treatment

## Abstract

**Simple Summary:**

Neoadjuvant immunochemotherapy (NICT) has demonstrated impressive short-term efficacy, with over half of patients experiencing significant tumor shrinkage and achieving major pathological responses (MPR). These findings highlight the pressing need for further investigation into strategies for organ preservation and radiotherapy adjustments in patients who achieve MPR. Our objective was to utilize non-invasive and accessible clinical assessments to predict pathological response before surgery. By employing enhanced CT scans, esophagograms, and esophagoscopy before and after neoadjuvant treatment, we collected objective and quantitative parameters that reflected the dynamic shrinkage of tumors. Subsequently, we constructed prediction models for pathological response using multivariate logistic regression based on those dynamic parameters. These models accurately predicted pathologic complete response (pCR) (AUC 0.879) and MPR (AUC 0.912) of the primary tumor after neoadjuvant immunochemotherapy. This advancement may significantly aid informed decision-making in patient management.

**Abstract:**

To develop accurate and accessible prediction methods for assessing pathologic response following NICT prior to surgery, we conducted a retrospective study including 137 patients with esophageal squamous cell carcinoma (ESCC) who underwent surgery after two cycles of NICT between January 2019 and March 2022 at our center. We collected clinical parameters to evaluate the dynamic changes in the primary tumor. Univariate and multivariate analyses were performed to determine the correlations between these parameters and the pathologic response of the primary tumor. Subsequently, we constructed prediction models for pCR and MPR using multivariate logistic regression. The MPR prediction Model 2 was internally validated using bootstrapping and externally validated using an independent cohort from our center. The univariate logistic analysis revealed significant differences in clinical parameters reflecting tumor regression among patients with varying pathologic responses. The clinical models based on these assessments demonstrated excellent predictive performance, with the training cohort achieving a C-index of 0.879 for pCR and 0.912 for MPR, while the testing cohort also achieved a C-index of 0.912 for MPR. Notably, the MPR prediction Model 2, with a threshold cut-off of 0.74, exhibited 92.7% specificity and greater than 70% sensitivity, indicating a low rate of underestimating residual tumors. In conclusion, our study demonstrated the high accuracy of clinical assessment-based models in pathologic response prediction, aiding in decision-making regarding organ preservation and radiotherapy adjustments after induction immunochemotherapy.

## 1. Introduction

Immunotherapy combined with chemotherapy has been recommended as the first-line treatment for advanced esophageal cancer patients based on multiple phase III trials, including Keynote-590, ESCORT-1st and CheckMate 648 [1,2,3]. The remarkable overall response rate of nearly 70% observed in the ESCORT-1st trial has encouraged investigators to explore the value of immunochemotherapy in the neoadjuvant phase. Impressively, neoadjuvant immunochemotherapy (NICT) displays extraordinary short-term efficacy, with a pooled 53.5% major pathologic response (MPR) rate and a 33.8% pathologic complete response (pCR) rate [4,5]. However, 40–50% of patients experience postoperative complications, including around 10% that suffer major postoperative complications [6,7,8]. Therefore, organ-saving strategies such as active surveillance or radiotherapy may be options for patients who achieve pCR or even MPR after neoadjuvant treatment. Moreover, several studies have demonstrated that higher radiotherapy doses and larger target volumes are correlated with adverse events and lymphocytopenia of radiotherapy [9,10]. For those who receive radiotherapy rather than surgery after immunochemotherapy induction due to physical condition or unwillingness to surgery, the dose prescriptions and target volume should be adjusted considering significant tumor volume shrinkage and less than 10% tumor residues in more than 50% of patients. Therefore, an accurate approach to evaluate patients’ responses after immunochemotherapy is a critical unmet need for future studies to investigate personalized treatment plans.

Several retrospective and prospective studies have investigated the accuracy of detecting residue disease by using different clinical assessments after neoadjuvant chemoradiotherapy (NCRT) [11,12,13,14]. The preSANO trial illustrated that an optimal approach to clinical response evaluation after NCRT might involve a combination of multiple endoscopic examinations and PET-CT scans [11]. Additionally, the ongoing multicenter observational study (PRIDE trial) aims to develop a multimodal model to detect residual disease by integrating magnetic resonance imaging (MRI), 18F-FDG PET/CT, along with the addition of endoscopic assessments [15]. These studies underscored the potential of amalgamating multiple clinical assessments to enhance the prediction of pathologic outcomes.

On the other hand, the effectiveness of response evaluation by clinical assessments is unclear in patients treated with NICT due to the different regression patterns and lower incidence of esophagitis [6,11]. A prospective clinical trial demonstrated the strong predictive ability of paired PET/CT before and after NICT in primary tumor pCR using the combination of multiple metabolic parameters (AUC 0.888) [16]. Nonetheless, the limited accessibility and costliness of PET-CT warrant the exploration of accessible and affordable diagnostic tests for pathologic response prediction. In the NICE trial, a moderate positive correlation (r = 0.600) was observed between CT-measured reduction in longest lesion diameter and pathologic regression rate [7]. This suggests that incorporating parameters evaluating the dynamic tumor regression based on paired diagnostic tests conducted before and after NICT could enhance the prediction of pathologic outcomes before surgery. Our study aims to investigate whether integrating multiple diagnostic tests, including CT, esophagogram and esophagoscopy, is sufficient for evaluating pathologic primary tumor response, aiding in decision-making for organ preservation or radiotherapy adjustments.

## 2. Materials and Methods

### 2.1. Patients

This retrospective study was approved by the ethics committee of Shanghai Chest Hospital. Thoracic esophageal cancer patients with clinical stage T1-2N+, T3-4N0-3, M0 or M1 lymph node metastasis (confined to the supraclavicular lymph nodes), according to the AJCC 8th edition, who underwent surgery after two cycles of neoadjuvant immunochemotherapy from January 2019 to March 2022, were included in this cohort. Patients were required to have pathologically confirmed esophageal squamous cell carcinoma (ESCC) and complete clinical assessments, including contrast-enhanced CT and esophagogram, performed three weeks before neoadjuvant treatment and surgery. Radical resections were performed 4–8 weeks after neoadjuvant treatment, and at least 12 lymph nodes were removed. Patients were excluded if they had (1) other uncontrolled malignant tumors, (2) non-squamous cell carcinoma or contained non-squamous cell carcinoma components, (3) multi-segmental esophageal cancer, (4) clinical stage T1-2N0M0 (5) received radiotherapy before surgery.

Another cohort for model validation was also retrospectively collected, including patients who underwent neoadjuvant immunochemotherapy and radical resection from April 2022 to March 2023 in Shanghai Chest Hospital. Patients were required to have CT, esophagogram and esophagoscope at baseline and before surgery. Other inclusion and exclusion criteria were the same as mentioned above.

### 2.2. Neoadjuvant Immunochemotherapy and Surgery

Patients were planned to receive 2 cycles of neoadjuvant immunochemotherapy. PD-1 monoclonal antibodies for immunotherapy, such as camrelizumab, Keytruda, tislelizumab, and sintilimab, were administered at a standard dose of 200 mg every 3 weeks for 2 cycles. Standard chemotherapy regimens consisted of platinum-based two drugs and defined as follows: (1) NICE regimen (qw): two cycles of nab-paclitaxel 100 mg/m^2^ (day 1, 8, 15) + carboplatin AUC 5 mg/mL/min (day 1) with a 21-day interval [7], and (2) Other regimens (q3w): two cycles of nab-paclitaxel 260 mg/m^2^ (day 1) or paclitaxel 135–175 mg/m^2^ (day 1) + carboplatin AUC 5 mg/mL/min (day 1), cisplatin 75 mg/m^2^ (day 1) or nedaplatin 75 mg/m^2^ (day 1) with a 21-day interval [17,18,19]. Reductions in chemotherapy agent dosages, following the guidelines of two dosage levels, were allowed for cases of severe (at least grade 3) febrile neutropenia, neutropenia, anemia, or thrombocytopenia. Dose reduction was classified as a decrease in chemotherapy dose ≥15% relative to the standard for ≥1 myelosuppressive agent or immunotherapy interruption due to adverse events in any given cycle. A dose delay was classified as a delay of ≥7 days from the standard regimen in any given cycle. Curative surgical resection was carried out within a window of 4–8 weeks following the completion of NICT. Esophagectomy, combined with a two-field lymphadenectomy, was conducted using one of the following techniques: McKeown or Ivor Lewis, depending on tumor location and the surgeon’s choices. Minimally invasive approaches, including thoracoscopic and robotic-assisted esophagectomy, were routinely employed, although open surgery was performed in select cases. In cases where cervical lymph node metastasis was suspected, a three-field lymphadenectomy was performed.

### 2.3. Pathologic Assessment

Pathologic specimens from each patient were evaluated by an experienced pathologist and reviewed by a senior pathologist specialized in esophageal diseases. Tumor regression grade (TRG) was classified into four categories according to the Chirieac grading system [20]: TRG1 represented no histologically identifiable residual carcinoma, TRG2 indicated 1–10% residual carcinoma, TRG3 denoted 11–50% residual carcinoma, and TRG4 indicated greater than 50% residual tumor. Pathologic complete response was defined as the absence of viable tumor residual (TRG1) at the primary tumor site, while MPR was categorized as less than 10% tumor residual (TRG1+TRG2) at the primary site.

### 2.4. Parameters from CT Scans

Chest CT scans with a slice thickness of 5 mm, performed within 3 weeks before treatment and surgery, were utilized for gross tumor volume (GTV) delineation. GTV-pre, defined as the visible extent of the primary esophageal tumor before neoadjuvant therapy, was manually contoured on pre-treatment CT slices by an experienced radiation oncologist using MIM software. The boundaries of GTV-pre were determined based on multiple complementary tests, including PET-CT, esophagogram, and esophagoscopy. The corresponding pre-surgery CT images were then fused and registered to the pre-treatment CT using anatomical alignment to delineate GTV-post. GTV-post was adjusted laterally based on the esophagus boundary after tumor regression while maintaining the same cranial-caudal length. The normal esophagus, with a length equal to GTV-pre, was delineated at least 1 cm away from GTV-pre in the cranio-caudal direction on the pre-treatment CT. The volumes of GTV-pre, GTV-post, and normal esophagus were calculated using MIM software as the total volume of lesions with a density ranging from −150 to 150 Hounsfield units (HU) in the target area. The formulas used to calculate the shrinkage of the primary tumor volume were as follows:$$post/pre=\frac{V_{GTV-post}}{V_{GTV-pre}}\times 100$$
$$GTV-residual=\frac{V_{GTV-post}-V_{normal}}{V_{GTV-pre}-V_{noraml}}\times 100$$

To delineate GTV-PET-pre, a subset of 32 patients with available PET-CT images at baseline and before surgery was selected. The initial GTV-PET-pre was automatically contoured using MIM software based on continuous lesions with a standardized uptake value (SUV) greater than 2.5, identified from the maximum SUV at the esophagus [21,22]. Manual adjustments were made to the contour based on the boundaries of adjacent anatomical structures, including identifiable lymph nodes, heart, great vessels, and vertebras. The volume of GTV-PET-pre was calculated as the total volume of lesions with a density ranging from −150 to 150 Hounsfield units (HU) in the target area on CT.

### 2.5. Efficacy Assessment on Esophagogram

The efficacy of neoadjuvant treatment was evaluated based on four dimensions of the esophagogram, including the degree of esophageal stricture, the degree of esophageal stenosis, tumor shrinkage, and the smoothness of the esophageal wall. The response was categorized into four grades according to the RECIST criteria: Progressive Disease (PD), Stable Disease (SD), Partial Response (PR), and Complete Response (CR). Each grade was assigned a score of two points to facilitate assessment by evaluators based on their expertise. The specific details of the response evaluation criteria are provided in Appendix B.

To assess inter-observer agreement, two experienced radiation oncologists specialized in thoracic cancer independently evaluated paired esophagograms (at baseline and before surgery) of the first 30 patients, following the scoring table while blinded to the surgery results. The intraclass correlation coefficient (ICC) was calculated for the score of each dimension as well as the total score. Subsequently, the esophagograms of patients in this cohort were retrospectively assessed by one of the radiation oncologists who was unaware of the pathological outcomes.

### 2.6. Endoscopic Response Evaluation

The endoscopic response to neoadjuvant immunochemotherapy was assessed by two experienced endoscopists. The endoscopy findings and macroscopic images were retrospectively reviewed to determine the response. The endoscopic response was categorized into two main groups based on the presence or absence of tumor residue. Residual disease was defined as the presence of a definite residual tumor, thickening of the esophageal wall, or the existence of small nodules. On the other hand, the complete disappearance of the tumor with replacement by scarring or complete flattening of the esophageal wall was classified as no residual disease. It is important to note that the presence of granuloma-like elevations and healing ulcers were not considered tumor residue. 

### 2.7. Statistical Analysis

Numerical variables were compared using t-tests, and the results were presented as median and standard deviation (SD). Categorical variables were tested using the chi-square test and presented as the number and percentage. Odds ratios (ORs) for continuous variables were calculated using binary logistic regression and interpreted as the odds ratio for every 1-unit increase. Multicollinearity among variables from the same assessments was assessed using the variance inflation factor (VIF), with a VIF > 5 indicating the presence of multicollinearity. The interrater reliability of the assessments was evaluated by calculating the ICC using a two-way model and absolute agreement. The ICC values were interpreted according to Koo and Li’s guidelines: <0.5, poor reliability; 0.5–0.75, moderate reliability; 0.75–0.9, good reliability; >0.9, excellent reliability. The agreement between GTV-pre and automatically contoured GTV-PET-pre was tested using Passing–Bablok regression. The maximum Youden index of the receiver operating characteristic (ROC) curve was used to determine the cut-off for each regression model. Multivariate logistic regression models were constructed to predict pCR or MPR based on parameters identified as independent factors through multivariable logistic analysis. All statistical analyses were conducted using R (version 4.1.2), and a *p*-value below 0.05 was considered statistically significant.

## 3. Results

### 3.1. Patient Characteristic and Pathological Outcomes

A total of 137 patients who received neoadjuvant immunochemotherapy were included in this study (Figure 1). The clinical characteristics and pathological outcomes are shown in Table 1. The majority were male (83.9%), with a mean age of 64.5 years. Pathologic response assessment showed that 38.0%, 25.5%, 9.5%, and 27.0% of patients achieved TRG grades 1–4, respectively. Univariate analysis revealed no significant association between baseline patient characteristics and pathologic response, except for a higher likelihood of pCR in female patients (*p* = 0.047). In terms of neoadjuvant treatment, 14.6% of patients experienced dose reduction, and 21.9% had treatment delays. The percentage of patients who underwent above 15% dose reduction was higher in the pCR group but did not reach statistical difference (*p* = 0.051). Patients with a better pathologic response exhibited fewer resected lymph nodes and better post-therapy pathologic N stage, suggesting improved control of metastatic lymph nodes in well-responded primary tumors.

### 3.2. Accuracy and Reproducibility of Response Evaluation

To assess the accuracy of GTV volume delineation, we compared manually contoured GTV-pre with automatically contoured GTV-PET-pre in 32 patients using Passing–Bablok regression (refer to Appendix A). The regression line had a slope of 0.98 and a Pearson correlation coefficient of 0.937, indicating a high level of agreement between the two methods.

Additionally, the reliability of the esophagogram scoring system in assessing response was evaluated by calculating the inter-observer reproducibility using the ICC. Among the five parameters assessed (total score and scores of four dimensions), four showed good reliability (0.9 > ICC > 0.75), while the smoothness score exhibited moderate reliability (0.75 > ICC > 0.5) (refer to Appendix A). These findings highlight the good inter-observer reproducibility of response evaluation using the esophagogram scoring system.

### 3.3. The Performance of Clinical Assessments in Pathologic Response Evaluation

The correlation between parameters from clinical assessments and pathologic responses was analyzed through univariate logistic regression analysis (Table 2). ROC curves were used to evaluate the predictive performance of each parameter, and the corresponding area under the curve (AUC) values were calculated (Table 3).

Regarding CT-based response evaluation, all three parameters showed a significant correlation with pathologic response, regardless of grouping by pCR or MPR (*p*-values < 0.05, *t*-test). Among these parameters, the post/pre ratio demonstrated the highest predictive ability, with an AUC of 0.727 for pCR and 0.770 for MPR.

The esophagogram scoring system also exhibited a significant association with pathologic response. All five parameters showed a statistically significant correlation. The total esophagogram score had the best discriminative performance in predicting pCR (AUC 0.793), while the smoothness score was the most effective predictor for MPR (AUC 0.866).

Out of the 137 enrolled patients, only 111 underwent esophagoscopy after neoadjuvant treatment. The endoscopic response was categorized as residual disease or no residual disease. The specificity of endoscopy in discriminating MPR was high at 0.878, with a sensitivity of 0.671. For pCR prediction, the specificity and sensitivity were 0.714 and 0.780, respectively (presented in Table 2). 

### 3.4. Multivariable Logistic Regression Analysis and Model Development

Due to the presence of strong collinearity among parameters in the esophagogram scoring system, only the parameter with the highest AUC was selected for each multivariable logistic regression analysis. For patients who did not undergo endoscopy after neoadjuvant treatment, whether clinical characteristics and parameters from CT and esophagogram were independently associated with primary tumor response was determined by multivariate analysis in the 137-patient population (see Appendix A). Only the independent parameters with a *p*-value less than 0.05 were included in the model development process. The performance of Model 1 is shown in Table 4, with a C-index of 0.818 for pCR prediction and 0.871 for MPR prediction. Subsequently, endoscopic findings were incorporated into Model 1 to create Model 2 in 111 patients who underwent esophagoscope before surgery. Model 2 demonstrated significantly superior discriminative performance compared to Model 1, with a high C-index of 0.912 for MPR prediction and 0.879 for pCR (see Table 4, Figure 2A,B).

### 3.5. Clinical Trial Grouping Based on MPR Prediction Model 2

A Phase II clinical trial, aiming to adjust radiotherapy dose and target area for unresectable locally advanced ESCC patients after the induction of immunochemotherapy, will be conducted in our center soon. The responses will be evaluated using CT scans, esophagogram, and esophagoscopy after the induction of therapy and predicted using MPR prediction Model 2. To avoid misestimating non-MPR patients as MPR, an additional cut-off of 1.05 was set for Model 2, achieving a specificity of 0.927 and a sensitivity above 0.7. A nomogram based on MPR prediction Model 2 was developed and shown in Figure 2C, classifying individuals with a probability above 0.74 (cut-off 1.05) as MPR patients.

To validate the performance of Model 2, internal validation was conducted using 1000 bootstrapping resamples. The corrected specificity and sensitivity for the cut-off of 1.05 were 0.930 [95% CI: 0.842–1.000] and 0.708 [95% CI: 0.600–0.810], respectively. The corrected AUC was 0.914 [95% CI: 0.854–0.963]. The calibration curve demonstrated excellent concordance between predicted values and outcomes, with a slope of 1.000 and a Brier score of 0.111 (see Figure 2D). The decision curve indicated a net benefit of 37.8% at the 0.74 cut-off, meaning that approximately 40% of patients could receive more moderate treatment without increasing the risk of false MPR prediction (see Figure 2E).

A separate cohort from 2022.4 to 2023.3 in our center was retrospectively collected for external validation. The clinical characteristics and pathologic outcomes of the training and validation cohorts are presented in Appendix A. The C-index in the testing set was 0.912, and the specificity and sensitivity for the cut-off of 1.05 were 0.941 and 0.792, respectively (see Figure 3). The calibration curve and decision curve for the testing set are shown in Appendix A. Similar performances were observed in the 41-patient validation cohort, demonstrating the accuracy of Model 2 for MPR prediction.

## 4. Discussion

To the best of our knowledge, this is the first study to evaluate the value of multiple clinical assessments for detecting residual disease after NICT and develop predictive models for pCR and MPR of primary tumors. Focusing specifically on esophageal squamous cell carcinoma, we employed clinical, non-invasive and easily accessible assessments to derive objective and quantitative parameters that accurately reflect tumor dynamic shrinkage, which have the potential for easy implementation in other medical centers, enhancing their practicality and generalizability. Our assessment criteria encompass a range of measures derived from CT scans, including tumor volume reduction metrics such as GTV-post, post/pre, and GTV-residual. In addition, we incorporated scores from a comprehensive esophagogram-based grading system that evaluated treatment efficacy across four dimensions: stricture, stenosis, tumor shrinkage, and smoothness. We also categorized endoscopic esophageal findings into two groups based on the presence or absence of tumor residuals. These parameters exhibited significant differences across different pathologic responses, ultimately serving as independent predictors for both pCR and MPR. Moreover, our models based on those parameters demonstrate promising predictive accuracy. The pCR prediction model, based on post/pre endoscopic findings and total esophagogram score, achieved an AUC of 0.879, a performance comparable to the performance of PET-CT in detecting residual disease after immunochemotherapy [16]. In terms of MPR prediction, our Model 2, which integrated post/pre endoscopic findings and esophagogram smoothness score, exhibited impressive accuracy with an AUC of 0.912, correctly identifying over 70% of MPR patients while only missing 7% of patients with TRG3 and TRG4 tumors when using a cut-off probability of 0.74. The incidence of underestimating tumor residues in our model was even lower than that of bite-on-bite biopsy in the preSANO trial, which missed 10% of patients with TRG3 or TRG4 tumors [11].

Preliminary follow-up results of patients receiving radical resection after neoadjuvant immunochemotherapy showed that a major response of the primary tumor was correlated with better disease-free survival (DFS), while no DFS difference was observed between patients with complete and major responses of the primary tumor. Moreover, it also demonstrated the high consistency (70%) between the pathologic response of lymph nodes and primary tumor [23]. This suggests that the major pathologic response (MPR) of the primary tumor might represent the total response to induction immunotherapy and could be used to group patients for further differentiated treatment based on their response, necessitating a reconsideration of the best treatment option for these patients.

Radical resection is a crucial intervention for locally advanced disease after neoadjuvant therapy, but postoperative complications remain a concern. Previous NICT studies have reported complications in 40–50% of patients, with around 10% experiencing major complications [6,7,8]. Furthermore, patients who underwent esophagectomy experienced a significant decline in their health-related quality of life [24,25]. Radiotherapy has emerged as an organ-saving strategy with similar survival outcomes and reduced treatment-related morbidity for patients achieving clinically complete or remarkable responses in other neoadjuvant settings [26]. Both the JCOG0909 trial and a randomized phase II trial in China demonstrated comparable survival between neoadjuvant chemoradiotherapy followed by surgery and definitive radiotherapy in patients with a clinical complete or good partial response [27,28]. For induction chemotherapy, the CROC trial revealed the effectiveness of chemoradiation in locally advanced ESCC patients who responded positively to three DCF chemotherapy courses, resulting in an 89.8% one-year progression-free survival rate [29]. In our study, we found that clinical complete response on different clinical assessments highly overlapped with major pathologic response, and our MPR prediction model may serve as a criterion for further clinical trials to investigate whether radiotherapy after induction immunochemotherapy can lead to non-inferior survival in patients with major responses. 

The radiation dose and target area should also be reconsidered for patients who received radiotherapy after induction immunochemotherapy. Previous studies have demonstrated that higher prescribed doses are associated with increased radiation exposure to organs at risk, elevated treatment toxicity, and a greater likelihood of postoperative complications in patients undergoing salvage surgery [9,30,31]. Moreover, the mean radiation dose received by normal tissues has been found to be correlated with radiotherapy-induced lymphopenia in thoracic diseases, which can negatively impact the effectiveness of immunotherapy and patients’ overall survival [10,32]. Therefore, for a substantial proportion of patients with locally advanced disease who almost achieve clinically complete tumor response and have less than 10% tumor residues after immunochemotherapy, it is plausible that a lower radiation dose could be adequate for controlling the subclinical residual disease while minimizing treatment-related side effects and protecting circulating lymphocytes [33,34]. Additionally, the inclusion of elective nodal irradiation should be carefully evaluated for patients with major pathologic responses, considering the lower likelihood of lymph node metastasis and the role of lymph nodes in antigen presentation and immune responses for immunotherapy [35,36]. Further investigation is needed to assess the impact of radiotherapy adjustments on survival outcomes, and the development of an accurate response prediction model aligned with major pathologic response is crucial for guiding individualized treatment and conducting clinical trials. 

In our study, we utilized the ratio of tumor volume before and after NICT to assess tumor response, which proved to be effective in predicting pCR (AUC 0.727, cut-off value 43.1) and MPR (AUC 0.770, cut-off value 50.4) of primary tumors. This approach differs from the NICE trial, which utilized the longest lesion diameter (measured on CT) to evaluate clinical tumor regression [7]. Measuring lesion length after neoadjuvant treatment relies on the subjective judgment and experience of the clinician, resulting in a low ICC of 0.550 [37]. Therefore, we chose tumor volume as our parameter and assumed it remained unchanged in the craniocaudal dimension to reduce observer bias based on previous studies in the NCRT settings [38]. The accuracy of tumor volume shrinkage in detecting residual disease was comparable in patients treated with NICT when compared to the previous meta-analysis of NCRT. The specificity and sensitivity for predicting pCR were approximately 0.69–0.93 and 0.56–0.77 in NCRT patients, according to a previous study, and 0.80 and 0.58 in our cohort of NICT patients [38,39,40]. However, the cut-off value for tumor shrinkage was nearly 50% for patients treated with NICT, which was higher than the cut-off value of 25% for patients treated with NCRT [39]. The thickening of the esophagus due to radiation-induced esophagitis might be the reason for less tumor volume shrinkage and a lower cut-off value in NCRT patients [41,42].

In addition, we developed a scoring system for response evaluation by esophagogram after NICT based on two Chinese esophagogram evaluation systems [43,44]. We observed a noteworthy correlation between pathologic complete or major response and complete response as assessed by each dimension of the esophagogram. This finding contrasts with previous research indicating no correlation between major pathologic response and changes in tumor length or lumen width on esophagogram following neoadjuvant chemotherapy [45]. Nevertheless, we have incorporated the lesion area, rather than the length or width of a single dimension, into our evaluation system, which could potentially enhance the response assessment.

We employed a stricter criterion to exclude any residual disease based on endoscopic findings, resulting in high specificity (0.88) and moderate sensitivity (0.67) in predicting MPR. Previous neoadjuvant chemotherapy studies have demonstrated a strong correlation between endoscopic response according to post-neoadjuvant macroscopic findings and pathologic response, with higher percentages of histological responders in endoscopic responders (45−58%) compared to endoscopic non-responders (7–16%) [46,47,48]. While the specificity and sensitivity of endoscopic evaluation slightly differ from our study due to different evaluation criteria, these Japanese studies support our findings.

There are several limitations to our study. Firstly, it was a retrospective study conducted at a single center, and therefore, the performance of multiple assessments in response evaluation after immunochemotherapy needs to be validated in cohorts from different centers. Secondly, the evaluation of response by endoscopy may be influenced by the experience of the endoscopist. To address this, the use of an endoscopic evaluation model employing deep neural networks could be considered for clinical practice. Additionally, the value of bite-on-bite assessment in detecting residues should be explored in future studies. Thirdly, our study focused only on predicting the pathologic response of the primary tumor, and the regression of lymph nodes was not included. Combining parameters of lymph node dynamic shrinkage may improve the prediction of the ypN stage and status of each lymph node. Lastly, integrating pre-treatment pathologic sections and clinical parameters from multiple time points might enhance the predictive performance of our models.

## 5. Conclusions

Our study successfully developed models that accurately predicted pCR (AUC 0.879) and MPR (AUC 0.912) of the primary tumor after neoadjuvant immunochemotherapy based on dynamic clinical parameters. The MPR prediction Model 2 correctly identified over 70% of MPR patients with a particularly low probability of underestimating residual disease. It could serve as a response evaluation method for organ-saving decision making and radiotherapy adjustments.

## Figures and Tables

**Figure 1 cancers-15-04377-f001:**
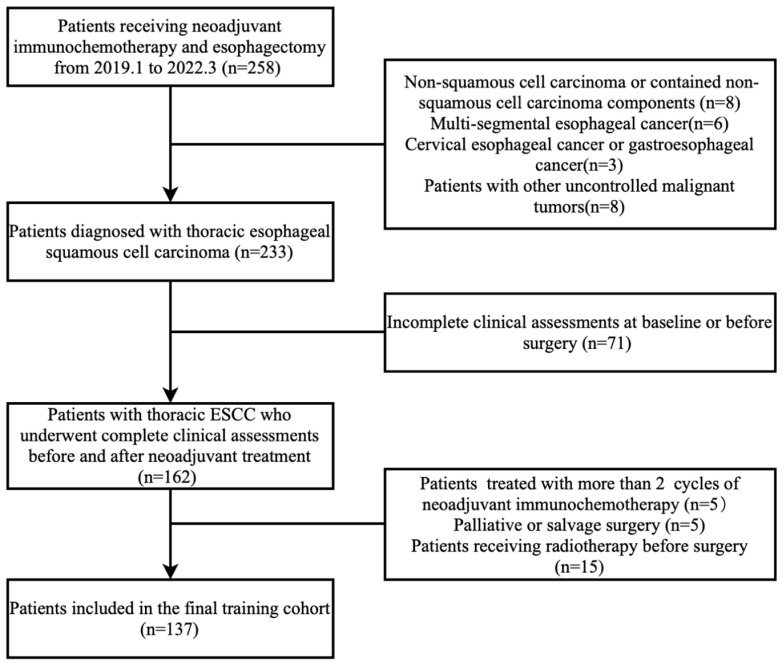
Flowchart of patient selection.

**Figure 2 cancers-15-04377-f002:**
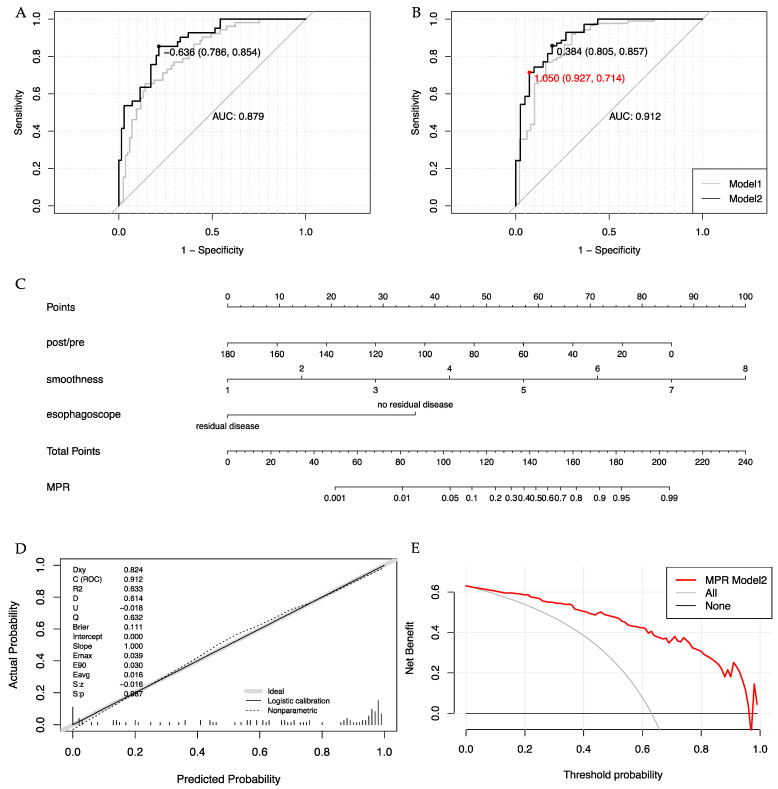
The performance of prediction models. ROC curve analysis of pCR prediction models (**A**) and MPR prediction models (**B**). The black dot represents the cut-off point with the maximum Youden index. The red dot represents the cut-off point of 1.05. Nomogram of MPR prediction Model 2 (**C**). The calibration plot (**D**) and the decision curve (**E**) of the MPR Model 2 in the 111-patient training cohort.

**Figure 3 cancers-15-04377-f003:**
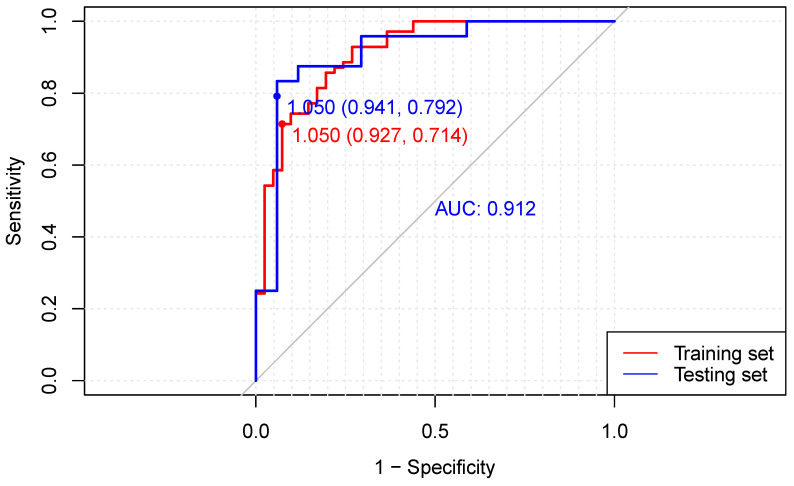
ROC curve analysis of MPR Model 2 in the 111-patient training cohort and 41-patient testing cohort. The red line represents the ROC curve for the training cohort, while the blue line represents the ROC curve for the testing cohort. Dots represent the cut-off point of 1.05, displaying their corresponding specificity and sensitivity values in both the training and testing sets.

**Table 1 cancers-15-04377-t001:** Clinical characteristics of patients according to tumor pathological response.

Characteristic	All	Non-pCR	pCR	*p*.Overall	Non-MPR	MPR	*p*.Overall
	N = 137	N = 85	N = 52		N = 50	N = 87	
Age	64.5 (6.93)	64.0 (7.07)	65.3 (6.69)	0.290	63.3 (6.89)	65.2 (6.91)	0.128
Sex:				0.047			0.460
Female	22 (16.1%)	9 (10.6%)	13 (25.0%)		6 (12.0%)	16 (18.4%)	
Male	115(83.9%)	76 (89.4%)	39 (75.0%)		44 (88.0%)	71 (81.6%)	
GTV-pre(cm^3^)	36.8 (21.9)	37.4 (20.6)	35.9 (23.9)	0.707	39.2 (21.2)	35.5 (22.2)	0.327
Tumor location:				0.362			0.655
Lower	72 (52.6%)	41 (48.2%)	31 (59.6%)		24 (48.0%)	48 (55.2%)	
Middle	43 (31.4%)	28 (32.9%)	15 (28.8%)		18 (36.0%)	25 (28.7%)	
Upper	22 (16.1%)	16 (18.8%)	6 (11.5%)		8 (16.0%)	14 (16.1%)	
Clinical-T:				0.109			0.229
T1-2	75 (54.7%)	42 (49.4%)	33 (63.5%)		24 (48.0%)	51 (58.6%)	
T3-4	62 (45.3%)	43 (50.6%)	19 (36.5%)		26 (52.0%)	36 (41.4%)	
Clinical-N:				0.120			0.251
N0	5 (3.65%)	1 (1.18%)	4 (7.69%)		0 (0.00%)	5 (5.75%)	
N1	25 (18.2%)	19 (22.4%)	6 (11.5%)		8 (16.0%)	17 (19.5%)	
N2	84 (61.3%)	51 (60.0%)	33 (63.5%)		31 (62.0%)	53 (60.9%)	
N3	23 (16.8%)	14 (16.5%)	9 (17.3%)		11 (22.0%)	12 (13.8%)	
Regimen				0.303			0.991
Others (q3w)	59 (43.1%)	40 (47.1%)	19 (36.5%)		21 (42.0%)	38 (43.7%)	
NICE(qw)	78 (56.9%)	45 (52.9%)	33 (63.5%)		29 (58.0%)	49 (56.3%)	
Dose reduction:				0.051			0.688
≤15%	117(85.4%)	77 (90.6%)	40 (76.9%)		44 (88.0%)	73 (83.9%)	
>15%	20 (14.6%)	8 (9.41%)	12 (23.1%)		6 (12.0%)	14 (16.1%)	
Dose delay:				0.368			0.813
<7 days	107(78.1%)	69 (81.2%)	38 (73.1%)		38 (76.0%)	69 (79.3%)	
≥7 days	30 (21.9%)	16 (18.8%)	14 (26.9%)		12 (24.0%)	18 (20.7%)	
s-LN group	13.9 (2.75)	14.0 (2.56)	13.8 (3.05)	0.762	14.1 (2.27)	13.8 (2.99)	0.492
s-LN number	31.3 (12.2)	32.8 (13.3)	28.9 (9.75)	0.049	33.8 (13.6)	29.9 (11.1)	0.084
yp-T:				<0.001			<0.001
T0	52 (38.0%)	0 (0.00%)	52(100%)		0 (0.00%)	52 (59.8%)	
T1	28 (20.4%)	28 (32.9%)	0 (0.00%)		12 (24.0%)	16 (18.4%)	
T2	14 (10.2%)	14 (16.5%)	0 (0.00%)		8 (16.0%)	6 (6.90%)	
T3	43 (31.4%)	43 (50.6%)	0 (0.00%)		30 (60.0%)	13 (14.9%)	
yp-N:				<0.001			0.001
N0	73 (53.3%)	34 (40.0%)	39 (75.0%)		18 (36.0%)	55 (63.2%)	
N1	34 (24.8%)	24 (28.2%)	10 (19.2%)		12 (24.0%)	22 (25.3%)	
N2	20 (14.6%)	18 (21.2%)	2 (3.85%)		12 (24.0%)	8 (9.20%)	
N3	10 (7.30%)	9 (10.6%)	1 (1.92%)		8 (16.0%)	2 (2.30%)	

pCR, pathological complete response; MPR, major pathological complete response; s-LN group, surgical lymph node group, was defined as the number of lymph nodes groups were removed from surgery; s-LN number, surgical lymph node number, was the number of lymph nodes were removed from surgery; NICE represented the regimen used in NICE trial.

**Table 2 cancers-15-04377-t002:** Univariate logistic regression associating parameters from clinical assessments with pathological response.

	Non-pCR	pCR	OR (95%CI)	*p*.Ratio	*p*.Overall	Non-MPR	MPR	OR (95%CI)	*p*.Ratio	*p*.Overall
GTV-post	21.8 (14.4)	14.1 (7.58)	0.93 [0.89;0.97]	0.001	<0.001	25.9 (16.3)	14.8 (7.79)	0.91 [0.87;0.95]	<0.001	<0.001
post/pre	62.4 (27.3)	44.6 (14.3)	0.95 [0.93;0.98]	<0.001	<0.001	71.3 (30.7)	46.6 (14.4)	0.95 [0.93;0.97]	<0.001	<0.001
GTV-residual	26.1 (79.2)	−15.64 (101)	0.99 [0.99;1.00]	0.026	0.013	39.4 (99.4)	−6.51 (80.4)	0.99 [0.98;1.00]	0.003	0.007
Stenosis	5.35 (1.88)	6.87 (1.44)	1.77 [1.36;2.30]	<0.001	<0.001	4.78 (1.85)	6.59 (1.54)	1.79 [1.42;2.26]	<0.001	<0.001
Dilation	5.80 (1.80)	7.21 (1.30)	1.83 [1.38;2.43]	<0.001	<0.001	5.18 (1.80)	7.00 (1.36)	1.95 [1.51;2.50]	<0.001	<0.001
Shrinkage	5.74 (1.58)	7.19 (0.91)	2.76 [1.82;4.18]	<0.001	<0.001	5.16 (1.63)	6.94 (1.02)	2.83 [1.94;4.12]	<0.001	<0.001
Smoothness	5.82 (1.66)	7.27 (0.79)	2.77 [1.80;4.25]	<0.001	<0.001	5.12 (1.64)	7.09 (0.94)	3.42 [2.23;5.24]	<0.001	<0.001
Esophagogram-total	22.7 (6.39)	28.5 (3.71)	1.27 [1.15;1.40]	<0.001	<0.001	20.2 (6.38)	27.6 (4.17)	1.28 [1.18;1.40]	<0.001	<0.001
Esophagoscope ^a^					<0.001					<0.001
No residual disease	20 (28.6%)	32 (78.0%)	Ref.	Ref.		5 (12.2%)	47 (67.1%)	Ref.	Ref.	
Residual disease	50 (71.4%)	9 (22.0%)	0.12 [0.04;0.28]	<0.001		36 (87.8%)	23 (32.9%)	0.07 [0.02;0.19]	<0.001	

pCR, pathological complete response; MPR, major pathological response. 95% CI: 95% confidence interval OR, odds ratio. The estimated pCR/MPR odds increase corresponds to the increment of the continuous variables by the following units: 1 mL for GTV-post, 1 unit for post/pre and GTV-residual, and 1 score for stenosis, dilation, shrinkage, smoothness, and esophagogram-total ^a^
*n* = 111, for patients who underwent esophagoscope before surgery

**Table 3 cancers-15-04377-t003:** ROC analyses of parameters from clinical assessments to predict pCR and MPR of primary tumor.

	pCR					MPR				
	AUC (95% CI)	CUT-OFF	Control vs. Case	Specificity	Sensitivity	AUC (95% CI)	CUT-OFF	Control vs. Case	Specificity	Sensitivity
GTV-post	0.682 [0.593;0.772]	14.0	>	0.671	0.615	0.742 [0.653;0.832]	17.5	>	0.700	0.713
post/pre	0.727 [0.641;0.813]	43.1	>	0.800	0.577	0.770 [0.684;0.856]	50.4	>	0.720	0.724
GTV-residual	0.686 [0.597;0.774]	15.5	>	0.541	0.808	0.754 [0.662;0.847]	13.1	>	0.720	0.724
Stenosis	0.749 [0.666;0.832]	6.5	<	0.682	0.750	0.782 [0.700;0.865]	6.5	<	0.860	0.678
Dilation	0.744 [0.663;0.826]	6.5	<	0.565	0.827	0.793 [0.717;0.870]	6.5	<	0.720	0.759
Shrinkage	0.792 [0.718;0.866]	6.5	<	0.671	0.885	0.854 [0.789;0.928]	6.5	<	0.780	0.805
Smoothness	0.776 [0.703;0.849]	6.5	<	0.576	0.865	0.866 [0.805;0.928]	6.5	<	0.792	0.805
Esophagogram-total	0.793 [0.716;0.869]	27.5	<	0.718	0.788	0.830 [0.754;0.906]	24.5	<	0.760	0.828
Model 1	0.818 [0.747;0.889]	0.155	<	0.859	0.654	0.871 [0.805;0.938]	0.195	<	0.700	0.920
Model 2 ^a^	0.879 [0.817;0.942]	−0.636	<	0.786	0.854	0.912 [0.857;0.968]	0.384	<	0.805	0.857

Cases were defined as pCR and MPR. Model 1 was based on parameters from CT and esophagogram; Model 2 was based on parameters from CT, esophagogram and esophagoscopy; pCR, pathological complete response; MPR, major pathological complete response; AUC, area under the ROC curve; ^a^ *n* = 111, for patients who underwent esophagoscope before surgery.

**Table 4 cancers-15-04377-t004:** Multivariate logistic regression models for pCR and MPR prediction based on parameters from clinical assessments and characteristics.

	pCR						mPR				
	OR	95%CI		*p*.Value	C-Index		OR	95%CI		*p*.Value	C-Index
Model 1: CT+esophagogram					0.818	Model 1: CT+esophagogram					0.871
post/pre	0.963	0.934	0.990	0.011		post/pre	0.964	0.935	0.990	0.011	
Esophagogram-total	1.237	1.123	1.386	<0.001		Smoothness	2.970	1.951	4.846	<0.001	
Model 2 ^a^: CT+esophagogram+esophagoscope					0.879	Model 2 ^a^: CT+esophagogram+esophagoscope					0.912
post/pre	0.963	0.927	0.995	0.034		post/pre	0.965	0.930	0.998	0.048	
Esophagogram-total	1.247	1.112	1.433	0.001		Smoothness	2.887	1.737	5.356	<0.001	
Esophagoscope (residual)	0.126	0.042	0.341	<0.001		Esophagoscope (residual)	0.068	0.015	0.233	<0.001	

pCR, pathological complete response of primary tumor; MPR, major pathological complete response; OR, odds ratio. The estimated pCR/MPR odds increase corresponds to the increment of the continuous variables by the following units: 1 unit for post/pre, and 1 score for smoothness and esophagogram-total ^a^ *n* = 111, for patients who underwent esophagoscope before surgery.

## Data Availability

The datasets used and analyzed during the current study are available from the corresponding author upon reasonable request.

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
