# Peer review of "Enhancing Prediction for Tumor Pathologic Response to Neoadjuvant Immunochemotherapy in Locally Advanced Esophageal Cancer by Dynamic Parameters from Clinical Assessments"

_cancers, 2023, doi:10.3390/cancers15174377_

Round 1

Reviewer 1 Report

Major comments

1.       Neoadjuvant immunochemotherapy: This study compares the NICT regimen with the NICE trial regimen and others (Table 1), but there is no mention of it at all in the “Materials and Methods” section.

2.       There is no description of surgery. Please add a description of the surgery.

3.       Preoperative endoscopy should be included in the mandatory "complete clinical assessments" of patient selection. Since the purpose of this study is to predict pCR or MPR from preoperative data, I do not think it is acceptable to incorporate pathology data into the preoperative data (endoscopic findings). Only the 111 cases in which endoscopy was performed should be analyzed.

Minor

1.       "the intraclass correlation coefficient (ICC)" is mentioned in several places. Spell it out only once.

2.       Figure 1:In the 5th column, "under" may be "underwent".

3.       In Table 1, the analysis results for the s-LN group and s-LN number sub-items are not listed.

The same abbreviation is repeatedly spelled out. Please check it.

Author Response

Thank you very much for taking the time to review the document. I greatly appreciate your feedback and have addressed each of the comments and suggestions in detail. Kindly refer to the attached document for my point-by-point response. 

Reviewer 2 Report

General comments:

1.Authors conducted this retrospective study focused on the prediction of tumor response. Previous several studies discussed the parameters on tumor response. The advantages of this study were the use of I/O for advanced esophageal cancer and the verified results of pathology.

Introduction:

2.You mentioned that that incorporating parameters evaluating the dynamic tumor regression based on paired diagnostic tests conducted before and after NICT could enhance the prediction of 76 pathologic outcomes before surgery. 

Can you explain if this would change follow-up or treatment recommendations?

3.From a conceptual point of view, more inspections do not necessarily mean more accurate results, and sometimes a very precise inspection may be representative.

Can you clarify on this point?

Materials and methods:

4.Were the same doses of the different immunotherapy drugs used in this study?

When should the drug dosage be adjusted or held?

5.You use the value of SUV 2.5 to calibrate the tumor. 

Are there references to support?

Results:

6.Table 1: 

You showed ypN1: 19.2%,  N2: 3.85%,  N3:19.2% in pCR group.

Can you explain?

Discussions:

7.In discussion, you discussed the prediction value of MPR and pCR. Taken together, what clinical parameters should we use to assess the treatment outcomes?

It will be more helpful to readers if these factors can be listed more systematically in the discussion so that readers can understand at a glance.

Conclusions:

8.In the conclusion, the key findings should be sorted out and summarized.

Author Response

Thank you very much for taking the time to review our manuscript. We greatly appreciate your feedback and have addressed each of the comments and suggestions in detail. Kindly refer to the attached document for my point-by-point response. 

Round 2

Reviewer 1 Report

The author answered most of my comments. I have no additional comments.